# Evaluation of a Single-Channel EEG-Based Sleep Staging Algorithm

**DOI:** 10.3390/ijerph19052845

**Published:** 2022-03-01

**Authors:** Shanguang Zhao, Fangfang Long, Xin Wei, Xiaoli Ni, Hui Wang, Bokun Wei

**Affiliations:** 1Centre for Sport and Exercise Sciences, Universiti Malaya, Kuala Lumpur 50603, Malaysia; s2013074@siswa.um.edu.my; 2Department of Psychology, Nanjing University, Nanjing 210093, China; dg1907008@smail.nju.edu.cn; 3Institute of Social Psychology, School of Humanities and Social Sciences, Xi’an Jiaotong University, Xi’an 710049, China; nixiaoli@xjtu.edu.cn; 4Department of the Psychology of Military Medicine, Air Force Medical University, Xi’an 710032, China; 5Xi’an Middle School of Shaanxi Province, Xi’an 710006, China; bokun@snnu.edu.cn

**Keywords:** EEG, sleep staging, support vector machine, decision tree, back propagation neural network, random forest

## Abstract

Sleep staging is the basis of sleep assessment and plays a crucial role in the early diagnosis and intervention of sleep disorders. Manual sleep staging by a specialist is time-consuming and is influenced by subjective factors. Moreover, some automatic sleep staging algorithms are complex and inaccurate. The paper proposes a single-channel EEG-based sleep staging method that provides reliable technical support for diagnosing sleep problems. In this study, 59 features were extracted from three aspects: time domain, frequency domain, and nonlinear indexes based on single-channel EEG data. Support vector machine, neural network, decision tree, and random forest classifier were used to classify sleep stages automatically. The results reveal that the random forest classifier has the best sleep staging performance among the four algorithms. The recognition rate of the Wake phase was the highest, at 92.13%, and that of the N1 phase was the lowest, at 73.46%, with an average accuracy of 83.61%. The embedded method was adopted for feature filtering. The results of sleep staging of the 11-dimensional features after filtering show that the random forest model achieved 83.51% staging accuracy under the condition of reduced feature dimensions, and the coincidence rate with the use of all features for sleep staging was 94.85%. Our study confirms the robustness of the random forest model in sleep staging, which also represents a high classification accuracy with appropriate classifier algorithms, even using single-channel EEG data. This study provides a new direction for the portability of clinical EEG monitoring.

## 1. Induction

Sleep is an extremely important physiological phenomenon for human beings, a process of restructuring the organism [1]. When people enter the sleep state, most of the physiological activities of the body are inert. At this time, the pituitary gland secretes more growth hormones and prohormones, promoting the adjustment and reorganization of cells and tissue repair, eliminating human fatigue, and preparing for human physiological activities when awake [2,3].

It is worth noting that sleep is not a single process and can be divided into different sleep periods depending on the depth of sleep [4,5]. Current research suggests that sleep staging is divided into three major stages distinguished by specific brain waves and their ratios: wake (W), no-rapid eyes movement (NREM), and rapid eye movement (REM) [6,7]. According to the Rechtstaffen and Kamp (R&K) guidelines, the NREM stage was further subdivided into four stages, 1, 2, 3, and 4 (also referred to as S1, S2, S3, and S4) [6]. In general, the standard R&K sleep is divided into six stages, namely W, S1, S2, S3, S4, and REM [7]. In 2007, the American Academy of Sleep Medicine (AASM) divided NREM into three phases consisting of NREM1 (N1), NREM 2 (N2), and NREM 3 (N3). Therefore, according to the AASM standard, the sleep epoch can be divided into five stages: W, N1, N2, N3, and REM. Accurate sleep staging is the foundation for understanding sleep mechanisms and the clinical diagnosis and treatment of sleep disorders.

Traditional sleep staging requires manual labeling by a professional physician based on Polysomnography (PSG) of subjects during sleep. Although manual labeling by experts enables accurate sleep staging, the disadvantages are cumbersome collection process and time-consuming manual labeling [8,9,10]. In addition, patients must wear special equipment and complete the PSG acquisition in the laboratory throughout the night [11,12]. The patient’s sleep efficiency is also affected by the discomfort of sleeping in an unfamiliar environment [13]. Based on these challenges, researchers have tried to develop scoring methods that automatically analyze sleep stages. In recent years, more and more studies have been conducted using machine learning algorithms for sleep staging based on features such as physiological signals such as electroencephalography (EEG), electrocardiogram (ECG), electrooculogram (EOG), electromyogram (EMG), and respiration [14,15,16]. Numerous studies have found that the EEG signals are considered the most important and commonly used signals in sleep staging analysis [17,18]. The authors of [19] used multiple EEG channels to sleep stages and obtained a high accuracy rate. However, equipment with multiple EEG channels limits the movement of participants and affects the portability and wearability of sleep quality assessment devices.

Automatic sleep staging based on single-channel EEG signals has become a research focus in this field. The authors of [20] extracted 39 features from the time domain, frequency domain, and nonlinear features of the EEG signal and obtained an accuracy of 85.7% using a support vector machine (SVM) algorithm for automatic classification of sleep. The authors of [21] performed staged sleep based on a random forest (RF) classifier, and the classifier could achieve 87.82% accuracy when the number of selected features was 136. The accuracy of sleep staging has largely relied on the type of classifier. Besides SVM and RF classifiers, K-nearest neighbors, linear discriminant analysis (LDA), and naive Bayes classifiers were also used to perform EEG sleep stage classification [9,22]. In addition to the different choices of classifiers, researchers also optimize the feature set selection to improve accuracy. This is because using more features means that more computational power is required, which also increases the complexity of the system. However, there is no uniform standard on feature optimization methods. Some studies directly chose feature selection methods, such as modified graph clustering ant colony optimization [21], to select the most optimal feature set from the feature pool for correlation and redundancy analysis. There are also studies that selected the feature with the highest weight as the most optimal feature set based on the weight of each feature [17]. It is also worth noting that electrode selection in single-channel-based automated staging is also an essential factor affecting the correct rate. Some studies have used F4-M1 channels [23], and others have used Pz-Oz channels, or Fpz-Cz channels, and staging based on prefrontal FP1 and FP2 channels [24,25,26,27]. Ghimatgar et al. revealed that the results of sleep stage staging using Fpz-Cz EEG signals were more accurate than other channels [21]. Additionally, most of the current tools based on a single-channel design use the Fpz-Cz channel [8,21]. In the present study, we also performed automatic staging of sleep based on EEG signals from the Fpz-Cz channel.

The difference from previous studies is that the current study used four classifiers, namely SVM, RF, backpropagation neural network (BPNN), and decision tree (DT), which have been applied in previous studies, to stage sleep. The optimal classifier was identified by comparing their classification accuracies in the same dataset. Due to the nonlinearity and non-stationarity character of EEG signals, it is not possible to fully reflect the signal characteristics by extracting features from only one dimension, resulting in poor classification results. Therefore, we used three types of parameters in this study: time domain, frequency domain, and nonlinear features, making the classifier obtain the optimal input. In addition, the optimization of the feature set was also the focus of this study. On the basis of retaining the original multi-dimensional features, we tried to use the embedded method to filter features for the establishment of the sleep staging model. The embedded method is a feature filtering method that uses machine learning algorithms and models to obtain the weight coefficients of each feature and then selects features based on the coefficients from largest to smallest [28]. If the feature set filtered by the embedded method can achieve the same staging accuracy, the cost of computing sleep staging will be reduced in practical applications.

In conclusion, this study aimed to find an optimal feature set that can perform automatic sleep staging based on single-channel EEG signals by optimizing classifier algorithms, feature extraction, and feature filtering, which provide a theoretical reference for the design of clinical portable devices.

## 2. Material and Method

### 2.1. Material

The sleep EEG data used in this study came from the Expanded Sleep-EDF (ES-EDF) database [29]. We selected 24 h EEG recordings (marked as SC) from 12 healthy subjects aged 21 to 34 years. The sample consisted of five males and seven females. The Fpz-Cz single-channel EEG signals were used in this study, and the sampling rate was 100 Hz. The 30 s EEG data (3000-point data) were defined as a sample. The sleep sample distribution selected is shown in Table 1. The sleep staging results were manually labeled by experts according to AASM standards. The staging accuracy of the proposed method was tested by labeling the results of experts.

### 2.2. Feature Extraction

As EEG signals have strong variability and are easily disturbed by other physiological signals and the external environment, it is necessary to preprocess the original data to eliminate the noise interference. This study used a finite impulse response (FIR) bandpass filter in the range of 0.5–45 Hz to denoise the original EEG data. In order to achieve the accurate staging of sleep, 57 features were extracted from the three aspects of time domain, frequency domain, and nonlinear features. Table 2 describes the characteristics of each signal. The features are described below.

#### 2.2.1. Time Domain Feature

The first to fourth moments (i.e., mean, variance, skewness, and kurtosis) are often used in statistical features of EEG signals. The calculation method is as follows:(1)$$X_{\mathrm{mean}}=\overset{ˉ}{S(n)}=\frac{1}{N}\overset{N}{\underset{n=1}{\sum }}\;S\left(n\right)$$
(2)$$s^{2}=\frac{1}{N}\overset{N}{\underset{n=1}{\sum }}\;X{\left(n\right)}^{2}$$
(3)$$\mathrm{skewness}=\frac{1}{Ns^{3}}\overset{N}{\underset{n=1}{\sum }}\;X{\left(n\right)}^{3}$$
(4)$$\mathrm{kurtosis}=\frac{1}{Ns^{4}}\overset{N}{\underset{n=1}{\sum }}\;X{\left(n\right)}^{4}-3$$

##### Zero Crossing Rate

The zero-crossing method is a systematic analysis method expressed in the waveform as the intersection of the waveform at that point with the horizontal midline of the waveform [30].

Calculating *X* (*i*) × *X* (*i* + 1) for *i* = 1, 2 …, *n* − 1 and counting the number *N_Z_* of *i* satisfying *X* (*i*) × *X* (*i* + 1) < 0, the zero-crossing rate can be defined as follows:(5)$$ZCR=\frac{N_{Z}}{N-1}$$

##### First-Order and Second-Order Difference and Its Normalization

Let *X*_1_(*n*) be a first-order difference of *X*(*n*) and *X*_2_(*n*) be a second-order difference of *X*(*n*); then, the following equations can be obtained [31,32,33]:(6)$$\begin{array}{c} X_{1}\left(n\right)=X\left(n+1\right)-X\left(n\right)\left(n=1,2,\cdots ,N-1\right)\\ X_{2}\left(n\right)=X\left(n+2\right)-X\left(n\right)\left(n=1,2,\cdots ,N-2\right) \end{array}$$

The mean value of the absolute value of the first-order difference:(7)$$\delta_{X}=\frac{1}{N-1}\overset{N-1}{\underset{n=1}{\sum }}\;\left|X_{1}\left(n\right)\right|$$

The mean value of the absolute value of the normalized first-order difference:(8)$${\tilde{\delta }}_{x}=\frac{\delta_{X}}{s}$$

The mean value of the absolute value of the second-order difference:(9)$$\gamma_{x}=\frac{1}{N-2}\overset{N-2}{\underset{n=1}{\sum }}\;\left|X_{2}\left(n\right)\right|$$

The mean value of the absolute value of the normalized second-order difference:(10)$${\tilde{\gamma }}_{X}=\frac{\gamma_{X}}{s}$$

##### Hjorth

The time domain Hjorth parameter, also known as the normalized slope descriptor, is a statistical function that can describe the instantaneous characteristics of EEG signals in both the time domain and frequency domain [34]. The Hjorth parameter consists of three descriptors: activity, mobility, and complexity. The activity represents the average power of the EEG signal, which is the variance. Mobility is used to measure the average frequency of EEG signals. Complexity is used to measure the bandwidth of an EEG signal.

Let *X*_1_(*n*) be a first-order difference of *X*(*n*) and *X*_2_(*n*) be a second-order difference of *X*(*n*); then, the following equations can be obtained:(11)$$\begin{array}{c} X_{1}\left(n\right)=X\left(n+1\right)-X\left(n\right)\left(n=1,2,\cdots ,N-1\right)\\ X_{2}\left(n\right)=X\left(n+2\right)-X\left(n\right)\left(n=1,2,\cdots ,N-2\right) \end{array}$$

Note that the first-order difference here is the same as *X*_1_(*n*) defined in the previous section. The mean values of the first- and second-order differences of *X*(*n*) are denoted as *μ_d_* and *μ_dd_*, respectively, which satisfy:(12)$$\mu_{d}=\frac{X\left(N\right)-X\left(1\right)}{N-1}$$
(13)$$\mu_{dd}=\frac{X\left(1\right)+X\left(N\right)-X\left(2\right)-X\left(N-1\right)}{N-2}$$

Then, their variances are denoted as *S_d_* and *S_dd_*, respectively, which satisfy:(14)$$s_{d}^{2}=\frac{1}{N-1}\overset{N-1}{\underset{n=1}{\sum }}\;{\left(X^{\prime }\left(n\right)-\mu_{d}\right)}^{2}$$
(15)$$s_{du}^{2}=\frac{1}{N-2}\overset{N-2}{\underset{n=1}{\sum }}\;{\left(X^{\prime }\left(n\right)-\mu_{\Delta }\right)}^{2}$$

On this basis, the activity, mobility, and complexity formulas are as follows:(16)$$\mathrm{Activity}=s^{2}$$
(17)$$\mathrm{Mobility}=\sqrt{\frac{s_{d}^{2}}{s^{2}}}=\frac{s_{d}}{s}$$
(18)$$\mathrm{Complexity}=\sqrt{\frac{\frac{s_{dd}s_{d}^{2}}{s_{d}^{2}}}{s^{2}}}=\frac{\frac{s_{d}}{s_{d}}}{\frac{s_{d}}{s}}=\frac{s\cdot s_{di}}{s_{d}^{2}}$$

#### 2.2.2. Frequency Domain Feature

Since EEG presents different rhythm distributions in different sleep stages, the filtered EEG signal was divided into seven frequency bands: low *δ*: 0.5–2 Hz; high *δ*: 1.2–4 Hz; *θ*: 4–8 Hz; *α*: 8–13 Hz; low *β*: 13–20 Hz; high *β*: 20–30 Hz; and *γ* wave: 30–45 Hz.

The energy can be obtained according to different frequency ranges. The specific calculation method is as follows:Total frequency band power:
$$P=\overset{N_{\mathrm{FT}}}{\underset{n=1}{\sum }}\;{\left(\frac{F\left(n\right)}{N_{\mathrm{FFT}}}\right)}^{2}$$
where *F*(*n*) is the results of the signal X(*n*) at frequency *n.*

2.*δ* band power:


$$P_{\delta }=\overset{N_{\mathrm{FE}}}{\underset{n=1}{\sum }}\;{\left(\frac{\left(F_{\mathrm{low}\text{-}\delta }\left(n\right)+F_{\mathrm{high}\text{-}\delta }\left(n\right)\right)}{N_{\mathrm{FFT}}}\right)}^{2}$$


3.*θ* band power:


$$P_{\theta }=\overset{N_{\mathrm{FFT}}}{\underset{n=1}{\sum }}\;{\left(\frac{F_{\theta }\left(n\right)}{N_{\mathrm{FFT}}}\right)}^{2}$$


4.*α* band power:


$$P_{\alpha }=\overset{N_{\mathrm{FFT}}}{\underset{n=1}{\sum }}\;{\left(\frac{F_{\alpha }\left(n\right)}{N_{\mathrm{FFT}}}\right)}^{2}$$


5.*β* band power:


$$P_{\beta }=\overset{N_{\mathrm{GT}}}{\underset{n=1}{\sum }}\;{\left(\frac{\left(F_{\mathrm{low}\text{-}\beta }\left(n\right)+F_{\mathrm{high}\text{-}\beta }\left(n\right)\right)}{N_{\mathrm{FFT}}}\right)}^{2}$$


6.*γ* band power:


$$P_{\gamma }=\overset{N_{\mathrm{FFT}}}{\underset{n=1}{\sum }}\;{\left(\frac{\left(F_{\mathrm{low}\text{-}y}\left(n\right)+F_{\mathrm{high}\text{-}\gamma }\left(n\right)\right)}{N_{\mathrm{FFT}}}\right)}^{2}$$


#### 2.2.3. Nonlinear Features

##### Fractal Dimension

The fractal dimension (FD) can be used to represent the complexity of the time domain signal. The Higuchi algorithm was used to calculate the fractal dimension feature FD of *X*(*n*), as described [35].

The calculation formula is as follows:(19)$$H_{m}\left(k\right)=\frac{N-1}{\left[\frac{N-m}{k}\right]k^{2}}\overset{\left[\frac{N-m}{k}\right]}{\underset{n=1}{\sum }}\;\left|X\left(m+nk\right)-X\left(m+\left(n-1\right)k\right)\right|$$
where [*x*] represents the maximum integer not exceeding *x*. The average value $\bar{H}\left(k\right)$ of *H_m_* (*k*) is calculated as follows:(20)$$\overset{ˉ}{H}\left(k\right)=\frac{1}{k}\overset{k}{\underset{m=1}{\sum }}\;H_{m}\left(k\right)$$

For different values of *K*, the calculated $\bar{H}\left(k\right)$ is different, but—log *k* is linearly related to log $\bar{H}\left(k\right)$. The least-square method is used to fit the line equation, in which the slope is the fractal dimension (FD) obtained.

Let *k_min_* = 1 and *k_max_* = [$\frac{N}{20}$], and calculate $\bar{H}\left(k\right)$ for all positive integers *K* (*k_min_* ≤ *K* ≤ *k_max_*), and further calculate the following:$$\begin{array}{c} \mu_{k}=-\frac{1}{k_{max}-k_{min}+1}\overset{k}{\underset{k=k_{kin}}{\sum }}\;\ln k\\ \mu_{H}=\frac{1}{k_{max}-k_{min}+1}\overset{k}{\underset{k=k}{\sum }}\;\ln\overset{ˉ}{H}\left(k\right) \end{array}$$

Then, the fractal dimension FD can be calculated by the following formula:$$\mathrm{FD}=\frac{\sum_{k=k_{min}}^{k_{max}}\;\left(\mu_{H}-\ln\overset{ˉ}{H}\left(k\right)\right)\left(\mu_{k}+\ln k\right)}{\sum_{k=k_{min}}^{k_{max}}\;{\left(\mu_{k}+\ln k\right)}^{2}}$$

##### Non-Stationary Index

The non-stationary index (NSI) measures the variation of the local mean over time. The signal is divided into m segments, the mean of each segment is calculated, and the NSI is defined as the standard deviation of these m means. A larger NSI indicates a larger oscillation of the local mean [36].

We used a large amount of experimental data as the basis, with the criterion of minimum variance and mean square error, and with the help of a ninth-order polynomial fit; after computational derivation, the stable value of NSI is best reflected as *m* = [0.15 × N]. Let *N = mq + r*, *q* being a positive integer and 0 ≤ *r* < m; then *X*(*n*) can be divided into *m* segments as follows:

If *r* > 0:$$X_{k}=\left\{X_{q\left(k-1\right)+1},\cdots ,X_{qk}\right\},k=1,\cdots ,m$$

If *r* = 0:$$\begin{array}{c} X_{k}=\left\{X_{\left(q+1\right)\left(k-1\right)+1},\cdots ,X_{\left(q+1\right)k}\right\},k=1,\cdots ,r;\\ X_{k}=\left\{X_{\left(q+1\right)r+q\left(k-r-1\right)+1},\cdots ,X_{qk+r}\right\},k=r+1,\cdots ,m_{0} \end{array}$$

Let *X_k_* be the average of the set *X_k_*, $\mu =\frac{1}{m}\overset{m}{\underset{k=1}{\sum }}\;{\overset{ˉ}{X}}_{k}$, and the NSI can be calculated according to the following equation [37]:$$\mathrm{NSI}=\sqrt{\frac{1}{m}\overset{m}{\underset{k=1}{\sum }}\;{\left({\overset{ˉ}{X}}_{k}-\mu \right)}^{2}}$$

##### Sample Entropy

The core of sample entropy lies in comparing the self-similarity of sequences by comparing the autocorrelation of equal-length subsequences in a sequence relative to the growth of subsequence length [38]. The calculation of sample entropy does not depend on the length of the data and has a better consistency.

For the signal *X*(*n*), the calculation method of sample entropy is as follows:

Expand *X*(*n*) into *N* − *m* + 1 subsequences of length *m*, denoted as *X_m_*_,1_, *X_m_*_,2_, …, *X*_*m*,*N*+*n*+1_, where *X_m,I_* = {*X*(*i*), *X*(*I* + 1), …, *X*(*i* + *m* − 1)} of 1 ≤ *I* ≤ *N* − *m* + 1.

Define the distance *d* between *X_m,i_* and *X_m,j_* as the absolute value of the maximum difference between the corresponding elements:$$d\left(X_{m,i},X_{m,j}\right)=\underset{k=0,\cdots ,m-1}{max}\;\left(\left|X\left(i+k\right)-X\left(j+k\right)\right|\right)$$

For a given *X_m,i_*, count the number of *j* (1 ≤ *j* ≤ *n* − *m* + 1, *j* ≠ *I*) whose distance between *X_m,I_* and *X_m,j_* does not exceed *r*, and write it as *B_i_*. For 1 ≤ *I* ≤ *N* − *m* + 1; the definition is the following:$$B_{i}^{m}\left(r\right)=\frac{1}{N-m-1}B_{i}$$

Define *B^m^*(*r*) as:$$B^{m}\left(r\right)=\frac{1}{N-m}\overset{N-m+1}{\underset{i=1}{\sum }}\;B_{i}^{m}\left(r\right)$$

Increase the dimension to *m* + 1, count *X_m,i_*, and count the number of *J* (1 ≤ *j* ≤ *N* − *m* + 1, *j* ≠ *I*) whose distance between *X_m,i_* and *X_m,j_* is not more than *r*, denoted as *A**_i_* and *A**_i_^m^*(*r*), defined as:$$A_{i}^{m}\left(r\right)=\frac{1}{N-m-1}A_{i}$$

Define *A^m^*(*r*) as:$$A^{m}\left(r\right)=\frac{1}{N-m}\overset{N-m}{\underset{i=1}{\sum }}\;A_{i}^{m}\left(r\right)$$

Thus, *B^m^*(*r*) is the probability that two sequences match *m* points under the similarity tolerance *r*, while *A^m^*(*r*) is the probability that two sequences match the *m* + 1 point. Sample entropy is defined as follow:$$\mathrm{SampEn}\;\left(m,r\right)=\underset{N\to \infty }{lim}\left\{-\ln\;\left[\frac{A^{m}\left(r\right)}{B^{m}\left(r\right)}\right]\right\}$$

When *N* is a finite value, it can be calculated by the following formula:$$\mathrm{SampEn}\left(m,r,N\right)=-\ln\left[\frac{A^{m}\left(r\right)}{B^{m}\left(r\right)}\right]$$

Usually choose *m* = 2 or *m* = 3; *r* = 0.2 *s*; and *s* is the standard deviation of *X*(*n*) [39].

### 2.3. Rank-Based Feature Selection Method

To simplify the computation process and improve the portability of the algorithm, we performed feature screening on the features extracted in Section 2.2. The embedded method uses machine learning algorithms and models to obtain the weight coefficients of each feature and selects the features from the largest to the smallest according to the coefficients. Therefore, the study used the feature selection method based on the tree model to filter the features, and Table 3 shows the weight coefficients of each feature. In this study, features with feature weight coefficients greater than 0.02 were selected as the final set of classification features, so a total of 11 features was selected, including T6, T7, F2, F5, F6, F8, F9, F12, F19, F22, and N2.

### 2.4. Classification Models

In this study, four algorithms, namely the support vector machine (SVM), backpropagation neural network (BPNN), random forest (RF), and decision tree (DT) algorithms, were chosen to classify the extracted features, and the classification accuracy was obtained.

SVM is a robust classifier widely used in supervised classification problems [40]. Before using SVM classification, all features were converted into sequences 0–1 by the z-score standardization method. In this study, a linear function was selected as the kernel function, and the hyperparameters were tuned by grid search. The BP neural network algorithm is the most widely used neural network machine learning algorithm, which mainly contains an input layer, an implicit layer, and an output layer, and each layer is interconnected with the others for signaling through neural nodes [41]. Before classification using a BP neural network, all features are normalized in the range [0, 1] using the min–max normalization method. Since this study divided sleep into five periods, the number of nodes in the output layer was set to five, the number of nodes in the implicit layer was set to 20, the number of neural nodes in the input layer needed to be set according to the number of feature values in different sample sets, and the learning efficiency was set to 0.1. RF is an integrated algorithm consisting of multiple decision trees, and is one of the more common classification algorithms [42]. The decision trees in this algorithm are independent of each other, and the input sample set is analyzed and processed separately. The classification results of each tree are collated to obtain the final classification result. The Gini index measures the purity of the sample set, where the smaller the value, the lower the probability of misclassification of the sample. The DT algorithm is an inductive learning algorithm, a classification rule obtained by induction on a chaotic set of instances based on instances [43]. There are two steps to deal with the classification problem of the decision tree: first, the classification model of the decision tree is generated by a learning training set; second, the model is used to classify unknown types of samples. The C4.5 decision tree algorithm was applied in this study, and the splitting index was the information gain rate.

### 2.5. Validation of Classification Models

After the classifier design, a fair evaluation needs to estimate its performance over a large number of objects corresponding to a selected set of features and classifier designs. In this study, 20% of the samples (1940 samples) were randomly selected from the dataset as the test set, and the remaining samples were used for training. The model was trained on the training set using five-fold cross-validation, using 80% of the samples in each round as the training subset and the remaining 20% as the test subset.

After training with the model, there are four main categories when examining the prediction effect of the model: true positive, which means the prediction is positive and positive; fake positive, which means the prediction is positive but negative; true negative, which means the prediction is negative but negative; and fake negative, which means the prediction is negative but positive. Four metrics, namely accuracy, precision, recall, and f1-score, were used as the evaluation metrics of the classifier [44].

(1) Accuracy is the simplest index, consisting of the number of correctly predicted observations divided by the total number of observations:$$\mathrm{accuracy}=\frac{\mathrm{TP}+\mathrm{TN}}{\mathrm{TP}+\mathrm{FP}+\mathrm{TN}+\mathrm{FN}}$$

(2) The precision describes the proportion of true positives among the predicted positive samples:$$\mathrm{precision}=\frac{\mathrm{TP}}{\mathrm{TP}+\mathrm{FP}}$$

(3) Recall is the percentage of all actual positive samples that are predicted to be positive:$$\mathrm{recall}=\frac{\mathrm{TP}}{\mathrm{TP}+\mathrm{FN}}$$

(4) f1 is a more balanced index between precision and recall:$$f1=\frac{\mathrm{TP}}{\mathrm{TP}+\frac{\mathrm{FN}+\mathrm{FP}}{2}}$$

## 3. Results

### 3.1. SVM Model: Results and Evaluation

Automatic staging of sleep EEG data was carried out using the SVM model. All 57-dimensional features were selected. After the model parameters were adjusted, the model with “C = 1.3, *γ* = 0.03” was selected for testing. The results show that the recognition rate of phase W was the highest, and that of phase N1 was the lowest, with an average accuracy of 81.86%, as shown in Table 4. The corresponding confusion matrix is shown in Figure 1. It can be seen from Figure 1 that the REM and N1 stages were most likely to be confused. The wrong predictions of the N3 stage are mainly concentrated in the N2 stage; the wrong predictions of the N2 stage are scattered in the N3, N1, and REM stages; and the wrong predictions of the W stage are mainly concentrated in the N1 stage.

The expert manual staging results are visualized with the SVM model staging results in Figure 2. The real label represents the result of manual staging by experts, while the prediction label is the result of the SVM model. According to the results in the figure, the sleep staging labeled by experts is highly consistent with that obtained by the SVM model, as in only 12 of the 100 samples the predicted labels did not match the real ones.

### 3.2. BPNN Model: Results and Evaluation

A BPNN model was used for automatic staging of sleep EEG. All 57-dimensional features were selected. After model parameters were adjusted, two hidden layers with 18 neurons in each layer were selected for testing. As shown in Table 5, the average recognition rate of stage W was the highest at 90%, followed by 84% of stage N2. The recognition rate of the N3 and REM stages was close to 75%, and the lowest recognition rate of the N1 stage was 66%, with an average accuracy of 78.33%. The corresponding confusion matrix is shown in Figure 3. It can be seen that the two are most easily confused in the REM period and N1 period; the wrong prediction of the N3 period is mainly concentrated in the N2 period; the wrong prediction of the N2 period, N1 period, and REM period is more scattered, indicating that these three periods are easily confused with other periods, and the wrong prediction of the W period is mainly concentrated in the N1 period.

The expert manual staging results were visualized with the BP neural network model staging results, as shown in Figure 4. Due to the large number of samples in the test set, only 100 samples are selected for visualization. According to the results in the figure, the sleep staging labeled by experts was highly consistent with that obtained by the BP neural network model, in which for only 12 of the 100 samples the predicted labels did not match the real ones.

### 3.3. DT Model: Results and Evaluation

The DT model was used for automatic staging of sleep EEG, and all 57-dimensional features were selected. After adjusting the model parameters, a tree model with a depth of 11 and a minimum number of leaf node samples of 11 was selected for testing, and the results were as follows. The results revealed the highest recognition rate of 88% for the W period, followed by 87% for the N3 period, and the lowest recognition rate of 62% for the N1 period, with an average accuracy rate of 76.25% (Table 6). The corresponding confusion matrix is shown in Figure 5. It can be seen that the REM period and N1 period were the two most easily confused, but the distinction between these two periods and the N3 period was relatively high, and this model had a better effect in distinguishing deep sleep from light sleep; the false prediction of the N3 period was mainly concentrated in the N2 period; the false prediction of the N2 period was scattered in the other four periods, and the false prediction of the W period was mainly concentrated in the N1 period.

The expert manual staging results were visualized with the DT model staging results, as shown in Figure 6. According to the results in the figure, the sleep staging labeled by experts was highly consistent with that obtained by the DT model, in which for only 13 of the 100 samples the predicted labels did not match the real ones.

### 3.4. RF Model: Results and Evaluation

The RF model was used for automatic staging of sleep EEG, and all 57-dimensional features were selected. After the model parameters were adjusted, a tree model with a random forest size of 100 trees, a depth of 22 per tree, and a minimum number of leaf node samples of 5 was selected for testing, with the following results. As can be seen from Table 7, the recognition rate of the W stage was the highest at 92%, followed by 91% (N3). The recognition rate of N2 and REM was about 80%. The lowest recognition rate of phase N1 was 73%, and the average accuracy of the five sleep stages was 83.61%. The corresponding confusion matrix is shown in Figure 7. The two most easily confused were the REM and N1 periods; the erroneous prediction of the N3 phase was mainly concentrated in the N2 phase, with a small number predicted as the W phase, which was caused by the low frequency and high amplitude characteristics of the waveform of N3, causing the model to misclassify it as EOG and thus predict it as the W phase; the wrong prediction of the N2 period was scattered over the N3, N1, and REM periods; the wrong prediction of the W period was mainly concentrated in the N1 period.

The expert manual staging results are visualized with the RF model staging results, as shown in Figure 8. According to the results in the figure, the sleep staging labeled by experts is highly consistent with that obtained by the RF model, in which for only 12 of the 100 samples the predicted labels did not match the real ones.

### 3.5. Comparison of the Results of Four Models before and after Feature Screening

The 11-dimensional features after feature selection were input into four machine learning models. The accuracy of the obtained models was compared with the accuracy of all features, as shown in Table 8. The results indicate that the RF model had better sleep staging than the other three models, with the highest recognition rate of 92.13% for stage W and the lowest recognition rate of 73.46% for stage N1, with an average accuracy of 83.56%. The results of sleep staging using the 11-dimensional features agreed with the results of sleep staging using all features at 94.85%.

## 4. Discussion

In this study, based on EEG signals of the Fpz-Cz-channel, a total of 57 features were extracted from three dimensions: time domain, frequency domain, and nonlinear parameters. Then, four classifiers, namely SVM, BPNN, DT, and RF, were used for automatic sleep staging. The results show that the four classifiers have consistent results, that is, the highest recognition rate for the W phase and the lowest recognition rate for the N1 phase. The RF model exhibits the highest recognition accuracy among the four classifiers, followed by SVM, BPNN, and DT.

We have sorted out previous studies regarding sleep staging, feature number, classifier, single-channel name and accuracy, and kappa coefficient. Our study has three advantages over previous studies. First, we used the Fpz-Cz channel EEG data with the best sleep staging effect [21]. Second, in terms of the feature number, we extracted 57 features from the time domain, frequency domain, and nonlinear parameters of the sleep EEG signal for machine learning. Additionally, we used the embedded method to optimize the features into 11 dimensions to explore their classification accuracy. Finally, although we did not use all classifiers in terms of classifier selection, we selected several classifiers that performed well in previous studies. Our results show that compared with other classifiers (Table 9), RF achieves higher accuracy and maintains robust classification results both with multidimensional features (57) and optimized feature sets (11), which is consistent with the results of other studies [9,45,46].

The performance of classifiers also relies heavily on the associated features. In this study, the embedded method was used to select features with feature weight coefficients greater than 0.02 as the final set of classification features. Among the 11 features, there are two features from the time domain, eight features from the frequency domain, and only one from the nonlinear domain. These findings indicated that frequency domain features accounted for a greater proportion of the automatic sleep staging, followed by time domain features, possibly because different sleep stages exhibited different frequency and energy characteristics. Studies have shown that *δ* and *θ* bands’ rhythm mainly existed in the N2 and N3 stages [49], while *α* and *β* bands’ rhythm was detected mostly in the REM, awakening, and N1 stages [47]. Moreover, the proportion of frequency domain features accounts for the highest proportion in the optimal feature set; thus, future studies may consider the accuracy of automatic staging explored by screening on frequency domain features.

In our study, regardless of the classifier algorithm used, the classification accuracy was extremely high for stage W, whereas the recognition accuracy was lower for stage N1. The stage characteristics of sleep staging may cause this. When in the W stage, the individual still has a fairly complete consciousness, and the prominent EEG signal is characterized by a mixture of alpha and beta waves with more pronounced EEG characteristics. The N1 is the transition period of the brain from the conscious state to the sleep state, where the alpha wave share gradually decreases, and theta waves begin to appear and gradually replace alpha waves, suggesting that the EEG signal changes significantly during this period [50]. Thus, the W phase with stable features is easier to identify than the N1 phase with more variable EEG signals.

It should be noted that previous studies have shown that an imbalance in the number of categories during staging will affect the final accuracy. This means that when the number of instances of one class in the training dataset far exceeds the number of instances of other classes, the results tend to classify the data into the larger category [51]. However, in this study, when using the staging data, the samples of both the W1 and N1 areas were 2029, and the differences in the number of samples for each classification were small, which could effectively avoid the problems caused by data distribution.

The boosting classification method was used in a previous study on the classification effect of single-channel EEG signals, which showed that after extracting signal features with ensemble empirical mode decomposition (EEMD), the classification accuracy of wake, REM, DS, and LS4 states could reach 92.66% [52]. In this study, although we did stage discrimination based on 57 features in the time domain, the frequency domain and nonlinear features, the classification accuracy with the RF and SVM classification models attained more than 80%, and RF achieved more than 90% classification accuracy for both the W and N3 stages. In addition, one point that surpasses previous studies in this study is that we used the embedded method to reduce the feature dimensions to 11; we still found better classification results under the RF model. The amount of data was reduced by feature screening, and the speed of computation and portability of the algorithm were improved. The results further confirm that single-channel EEG is an available monitoring technology, which will provide a new direction for the portability of clinical EEG monitoring.

The study has some disadvantages, which are mainly reflected in the results on sleep staging. First, the recognition rate of the REM and N1 phases was lower. Second, the wrong prediction of the W phase was mainly concentrated in the N1 phase. The main reasons for these two problems are as follows: the EEG of the REM and N1 stages are mainly low-voltage mixed frequency waves, and this study only extracted features based on EEG, resulting in the REM and N1 stages not being easily distinguished; for the second point, on the one hand, it is because there are slow eye movements in both the closed-eye W and N1 stages. On the other hand, during the transition from the W stage to the N1 stage, the experts’ interpretation is more subjective, making the accuracy of sleep staging results difficult to guarantee. Therefore, improving the recognition rate of the REM and N1 stages is still a direction to focus on in sleep staging research.

## Figures and Tables

**Figure 1 ijerph-19-02845-f001:**
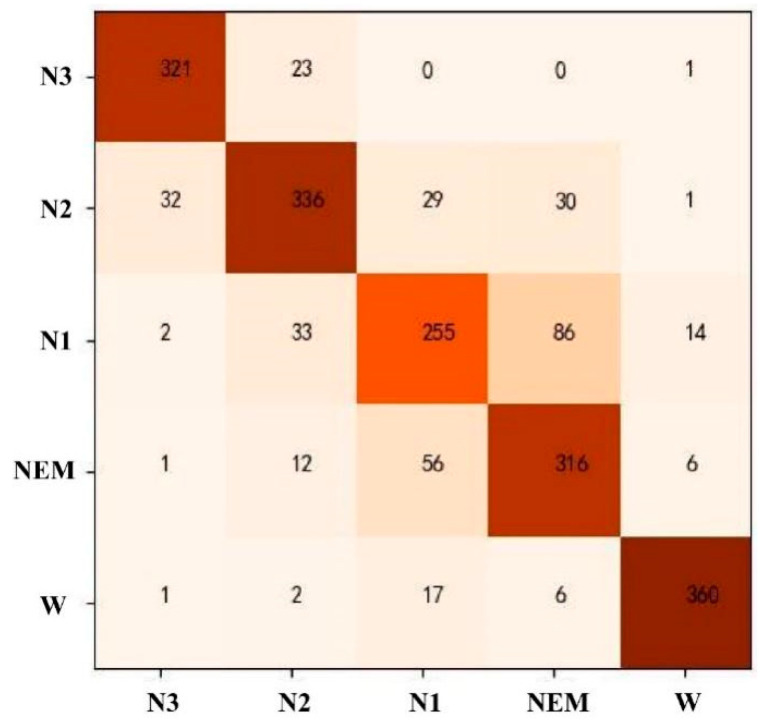
Confusion matrix of SVM model staging results.

**Figure 2 ijerph-19-02845-f002:**
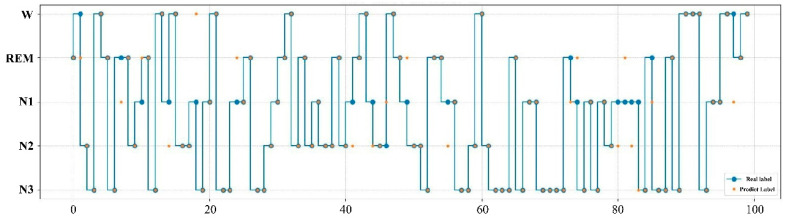
Comparison of expert manual staging results with SVM model staging results. The real label represents the result of manual staging by experts, while the prediction label is the result of the SVM model.

**Figure 3 ijerph-19-02845-f003:**
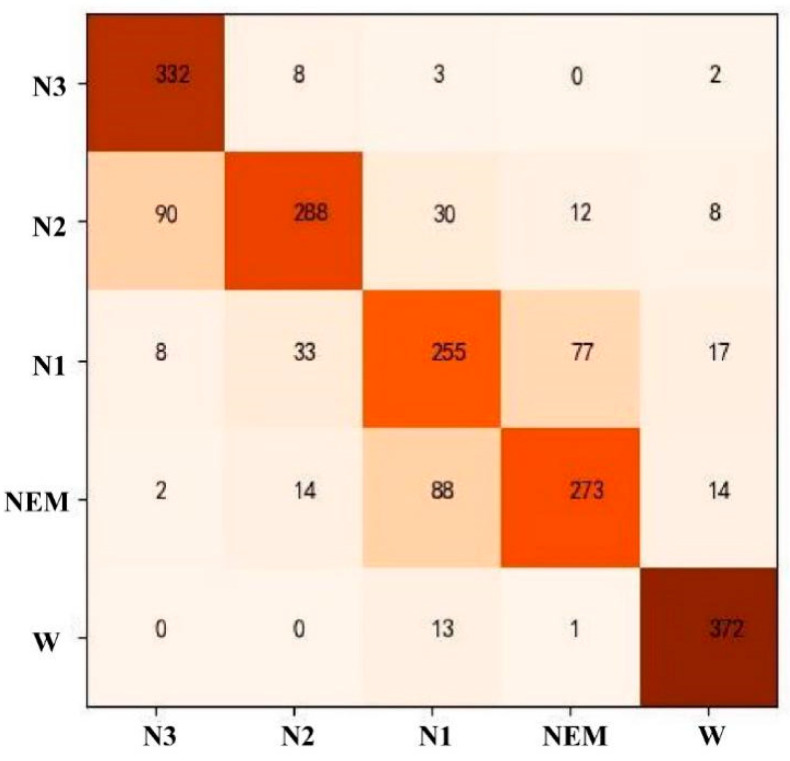
Confusion matrix of BPNN model staging results.

**Figure 4 ijerph-19-02845-f004:**
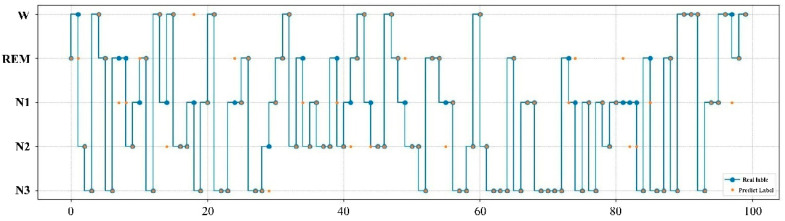
Comparison of expert manual staging results and BPNN model staging results. The real label represents the result of manual staging by experts, while the prediction label is the result of the BPNN model.

**Figure 5 ijerph-19-02845-f005:**
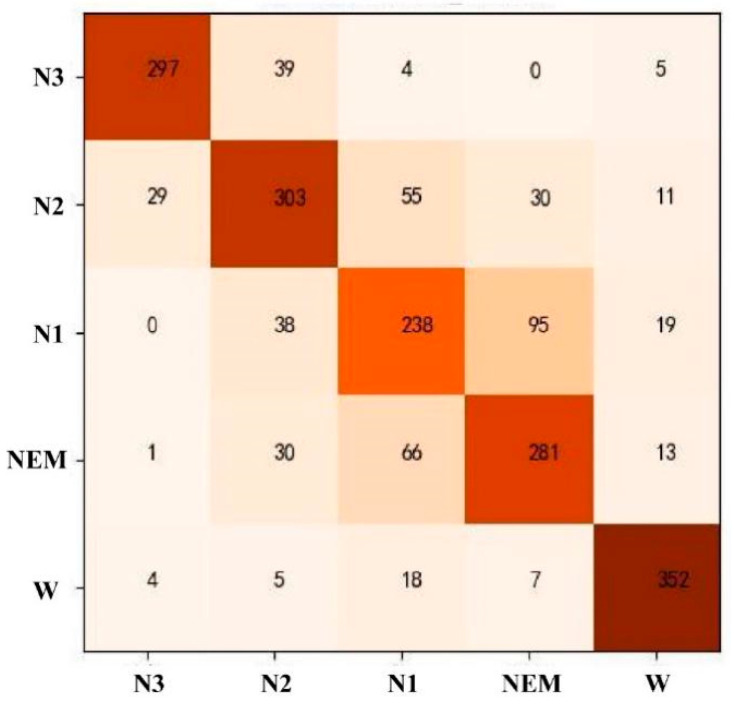
Confusion matrix of DT model staging results.

**Figure 6 ijerph-19-02845-f006:**
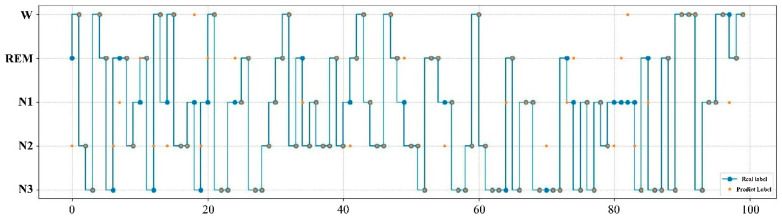
Comparison of expert manual staging results with DT model staging results. The real label represents the result of manual staging by experts, while the prediction label is the result of the DT model.

**Figure 7 ijerph-19-02845-f007:**
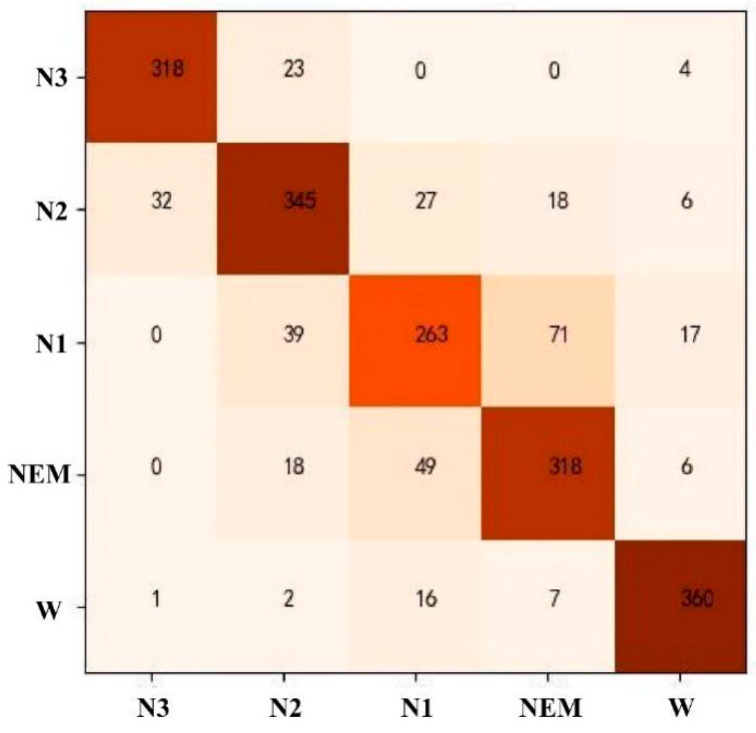
Confusion matrix of RF model staging results.

**Figure 8 ijerph-19-02845-f008:**
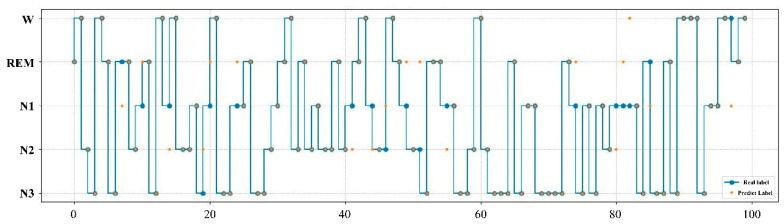
Comparison of expert manual staging results with RF model staging results. The real label represents the result of manual staging by experts, while the prediction label is the result of the RF model.

**Table 1 ijerph-19-02845-t001:** Sample distribution by stage.

Sleep Stages	Sample Number
W stage	2029
N1 stage	2029
N2 stage	2029
N3 stage	1671
REM stage	1938
Total	9696

**Table 2 ijerph-19-02845-t002:** EEG sleep staging characteristics.

Feature Symbol	Computational Method	Feature Symbol	Computational Method	Feature Symbol	Computational Method
T1	Amplitude	F5	E_6_ + E_7_	F24	(E_2_ + E_3_)/E_4_
T2	Mean Value	F6	E_8_	F25	(E_2_ + E_6_)/E_9_
T3	Variance	F7	E_2_/E_1_	F26	MPF
T4	SD	F8	E_3_/E_1_	F27	MPF-low-*δ*
T5	Median	F9	E_4_/E_1_	F28	MPF-high-*δ*
T6	Skewness	F10	E_5_/E_1_	F29	MPF-*θ*
T7	Kurtosis	F11	(E_6_ + E_7_)/E_1_	F30	MPF-*α*
T8	Maximum	F12	E_8_/E_1_	F31	MPF-*β*
T9	Minimum	F13	(E_4_ + E_5_)/E_1_	F32	MPF-*γ*
T10	ZCR	F14	E_5_/(E_6_ + E_7_)	F33	FV
T11	AFDN	F15	(E_4_ + E_5_)/(E_5_ + E_6_ + E_7_)	F34	FV-low-*δ*
T12	ASDN	F16	E_4_/(E_6_ + E_7_)	F35	FV-high-*δ*
T13	Activity	F17	E_3_/(E_4_ + E_5_)	F36	FV-*θ*
T14	Mobility	F18	E_4_/(E_3_ + E_5_)	F37	FV-*α*
T15	Complexity	F19	E_5_/(E_3_ + E_4_)	F38	FV-*β*
F1	E_1_	F20	E_2_/(E_3_ + E_9_)	F39	FV-*γ*
F2	E_2_ + E_3_	F21	E_5_/E_9_	N1	FD
F3	E_4_	F22	(E_6_ + E_7_)/E_9_	N2	NSI
F4	E_5_	F23	E_5_/E_4_	N3	E

T, time domain features; F, frequency domain features; N, non-stationary features. FD, fractal dimension; SE, sample entropy; ZCR, zero crossing rate; SD, standard deviation; E_1_, the total band power; E_2_, the low-frequency *δ*-band (0.5–2 Hz) power; E_3_, the high-frequency *δ*-band (1.2–4 Hz) power; E_4_, the *θ*-band (4–8 Hz) power; E_5_, the *α*-band (8–13 Hz) power; E_6_, the low *β*-band (13–20 Hz) power; E_7_, the high-frequency *β*-band (20–30 Hz) power; E_8_, the low-frequency *γ*-band (30–45 Hz) power; E_9_, the *δ* + *θ* + *α* + *β* + *γ* + *δ* band power.

**Table 3 ijerph-19-02845-t003:** Weight coefficients of EEG sleep staging features.

Feature Symbol	Weight Coefficient	Feature Symbol	Weight Coefficient	Feature Symbol	Weight Coefficient
T1	0.0030	F5	0.0394	F24	0.0012
T2	0.0006	F6	0.1459	F25	0.0053
T3	0.0051	F7	0.0038	F26	0.0011
T4	0.0122	F8	0.0281	F27	0.0028
T5	0.0111	F9	0.0457	F28	0.0007
T6	0.0513	F10	0.0038	F29	0.0011
T7	0.0201	F11	0.0106	F30	0.0013
T8	0.0015	F12	0.2043	F31	0.0007
T9	0.0066	F13	0.0024	F32	0.0006
T10	0.0086	F14	0.0142	F33	0.0007
T11	0.0013	F15	0.0124	F34	0.0037
T12	0.0049	F16	0.0110	F35	0.0014
T13	0.0101	F17	0.0105	F36	0.0006
T14	0.0048	F18	0.0104	F37	0.0010
T15	0.0050	F19	0.0540	F38	0.0005
F1	0.0148	F20	0.0008	F39	0.0006
F2	0.1049	F21	0.0031	N1	0.0045
F3	0.0056	F22	0.0301	N2	0.0470
F4	0.0125	F23	0.0077	N3	0.0032

**Table 4 ijerph-19-02845-t004:** Comparison of SVM model staging results for all features with expert manual staging results.

Sleep Stages	Training Samples	Test Samples	Correct Samples	Precision	Recall	f1-Score
N3	1326	345	321	0.8992	0.9304	0.9145
N2	1601	428	336	0.8276	0.7850	0.8058
N1	1639	390	255	0.7143	0.6538	0.6827
REM	1547	391	316	0.7215	0.8082	0.7624
W	1643	386	360	0.9424	0.9326	0.9375

Accuracy = 81.86%.

**Table 5 ijerph-19-02845-t005:** Comparison of BP model staging results for all features with expert manual staging results.

Sleep Stages	Training Samples	Test Samples	Correct Samples	Precision	Recall	f1-Score
N3	1326	345	321	0.7685	0.9623	0.8546
N2	1601	428	336	0.8397	0.6729	0.7471
N1	1639	390	255	0.6555	0.6538	0.6547
REM	1547	391	316	0.7521	0.6982	0.7241
W	1643	386	360	0.9007	0.9637	0.9312

Accuracy = 78.35%.

**Table 6 ijerph-19-02845-t006:** Comparison of decision tree model staging results for all features with expert manual staging results.

Sleep Stages	Training Samples	Test Samples	Correct Samples	Precision	Recall	f1-Score
N3	1326	345	321	0.8773	0.8609	0.8787
N2	1601	428	336	0.7301	0.7079	0.7189
N1	1639	390	255	0.6247	0.6103	0.6174
REM	1547	391	316	0.6804	0.7187	0.6990
W	1643	386	360	0.8800	0.9119	0.8957

Accuracy = 75.82%.

**Table 7 ijerph-19-02845-t007:** Comparison of random forest model staging results for all features with expert manual staging results.

Sleep Stages	Training Samples	Test Samples	Correct Samples	Precision	Recall	f1-Score
N3	1326	345	321	0.9171	0.9304	0.9237
N2	1601	428	336	0.8199	0.8294	0.8246
N1	1639	390	255	0.7346	0.7032	0.7032
REM	1547	391	316	0.7877	0.8015	0.8015
W	1643	386	360	0.9213	0.9404	0.9308

Accuracy = 83.56%.

**Table 8 ijerph-19-02845-t008:** Comparison of sleep staging accuracy of different models before and after feature screening (%).

Sleep Stages	SVM		BP		DT		RF	
	57	11	57	11	57	11	57	11
N3	89.92	88.67	76.85	89.89	89.73	88.48	91.71	92.11
N2	82.76	76.58	83.97	80.05	73.01	70.91	81.99	80.61
N1	71.43	70.50	65.55	63.27	62.47	61.42	73.46	72.85
REM	72.15	64.44	75.21	72.95	68.04	69.57	78.77	76.09
W	94.24	91.75	90.07	95.79	88.00	90.84	92.13	91.90
Total accuracy	81.86	77.99	78.35	79.59	75.82	75.88	83.56	82.53

**Table 9 ijerph-19-02845-t009:** The accuracy of sleep staging using single-channel EEG information.

Author/Year	Sleep Stages	Number of Features	Classifier	Channel of EEG	ACC (%) or KC
[20]	Five stages	39	SVM	C3-A2	ACC = 85.7%
[47]	Five stages	Multiscale entropy and autoregressive models	LDA	C3-A2	KC = 0.81
[48]	Five stages	9	SVM	Pz-Oz	ACC = 87.5% KC = 0.81
[9]	Five stages	The IMFs factor was set to 7 to obtain the optimal number of features	LDA, BPNN, SVM, k-NN, LS-SVM, Bagging, AdaBoost and Naïve Bayes	Pz–Oz	ACC (44.80–88.62%), AdaBoost algorithm has the highest accuracy of 88.62%
[45]	Five stages	10	K-NN, DT, RF, Multilayer perceptron and Naïve Bayes	Fpz-Cz (highest ACC), Cz-A1, C3-A2, Pz-Cz	ACC (71.80–89.74%) RF classifier had the highest accuracy of 89.74%
[21]	Five stages	136	RF	Fpz-Cz (highest ACC), Cz-A1, C3-A2, Pz-Cz	ACC = 87.82%
The present study	Five stages	57 and 11 (Embedded method feature optimization)	SVM, DT, RF and BPNN	Fpz-Cz	The ACC of 57 features (75.82–83.56%), RF with the highest The ACC of 11 features (75.88–82.53%), RF with the highest

Note: KC, kappa coefficient; ACC, accuracy; SVM, support vector machine; LS-SVM, Least Squares-support vector machine; DT, decision trees; RF, random forest; LDA, linear discriminant analysis; BPNN, backpropagation neural network; k-NN, k-nearest neighbor.

## Data Availability

The sleep EEG data used in this study came from the Expanded Sleep-EDF (ES-EDF) database.

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
