# Peer review of "Evaluation of a Single-Channel EEG-Based Sleep Staging Algorithm"

_ijerph, 2022, doi:10.3390/ijerph19052845_

Round 1

Reviewer 1 Report

This sentence: ”The patient's sleep efficiency is also affected by the discomfort of sleeping in 60 an unfamiliar environment” could be supported be the report showing different results in first and second night of the examination (see: 10.5664/jcsm.7036)

Author Response

On behalf of my co-authors, we thank you very much for giving us an opportunity to revise our manuscript, we appreciate editor and reviewers very much for their positive  comments and suggestions on our manuscript entitled “Evaluation of a single-channel EEG-based sleep staging algorithm”. (ID: ijerph-1498967).

We have added the literature you mentioned. 

Please see page 2, line 60 (in red text).

Reviewer 2 Report

The authors used a single channel EEG data to classify sleep stage and compared several machine learning algorithms in their performance. They first picked up about 10,000 epochs from open-sourse PSG data (Sleep-EDF Database Expanded from PhysioNet) and calculated 57 properties of EEG data of each epochs and used them for the machine learning.

Major problem of this article is insufficiency of description especially in methods. For example, there is no description how the authors picked up a limited number of epochs from a huge dataset. Followings are my comments.

  1. Sleep-EDF dataset has two different parts, ST and SC. Clarify which part was used.
  2. Sleep-EDF dataset has PSG data from different age groups, which show age-specific difference. The authors should clarify which age data are used and should discuss their results can be applied to all age group or only to a limited age group.
  3. The authors apparently used almost same number of epochs of W to REM stages as shown in Table 1. But the actual occurrence is different and apparently the authors picked this number on purpose. There is neither description of selection method nor the rationale for arbitrary selection of this evenly distributed datset.
  4. No explanation of E1 to E9 in Table 2. The authors used different font for E1, E2 in F1, F2 and F7. Do they mean the same thing?
  5. Fig 2, 4 … show hypnogram like plot. What are they? Are they artificially generated series of data? If so, explain how this series were produced. In addition, it is very hard to see the difference between real label and predicted label. Please change the color so that it can be distinguishable.
  6. In line 41 and 49, the word “cycle” is inappropriate. Sleep cycle means a series of NREM and REM sleep which consists of many epochs of different stages. They should be “epoch” or “stage”.

Reviewer 3 Report

This study compared various machine learning classification algorithms using a single-channel EEG for sleep staging, and concluded that the random forest with features including time domain, frequency domain, and nonlinear indices showed the highest accuracy which maintained although the number of features was compressed. This can give insight and interest to the readers of the journal.

1. If possible, I recommend to add a table including comparative accuracy of this research and the previous studies (listed in lines 72-98) which evaluated a single-channel EEG sleep staging.

2. Is there any advantages using the machine learning algorithms in this study over deep learning algorithms? Because there are several studies using deep learning algorithms for sleep staging recently.

Round 2

Reviewer 2 Report

As for the point #3, the authors misunderstood my question. My question was how they select the 9696 epochs (equal 4848min) from 12 PSG records. Since W, N1, N2 are 2029 epochs, I do not think they are accidentally same. It is unlikely to have the exactly same number of epochs of three different stages. I would like the authors to clarify this point.

Author Response

De'a'r

We apologize for the misunderstanding.

Data were available for 59 of 74 subjects in the SC dataset.  There are 19 people aged between 26 and 35, including 9 males and 10 females.  We extracted 1671-2029 epochs data from the available data, and excluded the epoch with more EEG artifacts caused by disturbances such as sweating and turning over. 80% as training set and 20% as test set. The 2029 occurrences of W, N1 and N2 respectively are purely coincidental.